# Quantized Variational Inference

**Amir Dib**
Université Paris-Saclay, CNRS, ENS Paris-Saclay, Centre Borelli, SNCF, ITNOVEM.
91190, Gif-sur-Yvette, France
amir.dib@ens-paris-saclay.fr

## Abstract

We present Quantized Variational Inference, a new algorithm for Evidence Lower Bound maximization. We show how Optimal Voronoi Tesselation produces variance free gradients for Evidence Lower Bound (ELBO) optimization at the cost of introducing asymptotically decaying bias. Subsequently, we propose a Richardson extrapolation type method to improve the asymptotic bound. We show that using the Quantized Variational Inference framework leads to fast convergence for both score function and the reparametrized gradient estimator at a comparable computational cost. Finally, we propose several experiments to assess the performance of our method and its limitations.

## 1 Introduction

Given data $y$ and latent variables $z$, we consider a model $p(y, z)$ representing our view of the studied phenomenon through the choice of $p(y|z)$ and $p(z)$. The goal of the Bayesian statistician is to find the best latent variable that fits the data, hence the likelihood $p(z|y)$. These quantities are linked by the bayes formula which gives that $p(z|y) = \frac{p(z)p(y|z)}{p(y)}$ where $p(y)$ is the prior predictive distribution (also named marginal distribution or normalizing factor) which is a constant. Given a variational distribution $q_\lambda$, the following decomposition can be obtained [36]

$$\log p(y) = \underbrace{\mathbb{E}_{z \sim q_\lambda} \left[ \log \frac{p(z, y)}{q_\lambda(z)} \right]}_{\text{ELBO}(\lambda)} + \underbrace{\text{KL}\big(q_\lambda(z) \| p(z|y)\big)}_{\text{KL-divergence}}. \tag{1}$$

It follows that maximazing the ELBO with respect to $q_\lambda$ leads to find the best approximation of $p(z|y)$ for the Kullback–Leibler (KL) divergence. Intuitively, this procedure minimizes the information loss subsequent to the replacement of the likelihood by $q_\lambda$ but other distances can be used [2].

The reason for the popularity of such techniques is due to the fact that finding a closed-form for $p(z|y)$ requires to evaluate the prior predictive distribution and thus to integrate over all latent variables which lead to intractable computation (except in the prior conjugate case) even for simple models [12]. A common approach is to use methods such as Gibbs Sampling, Monte Carlo Markov Chain or Hamilton Monte Carlo [3, 17, 6] which rely solely on the unnormalized posterior distribution (freeing us from the need to compute $p(y)$) and the ability to sample from the posterior. These methods are consistent but associated with heavy computation, high sensitivity to hyperparameters and potential slow to converge to the true target distribution. On the other hand, optimization techniques such as Variational Inference (VI) are generally cheaper to compute, tend to converge faster but are often a crude estimate of the true posterior distribution. Recent work proposes to combine these two strategies to allow for an explicit choice between accuracy and computational time [35].

Thanks to approaches such as Black Box Variational Inference (BBVI) [31, 20] (which opens the possibility of the generic use of VI), Automatic Variational Inference (AVI) [22] and modern

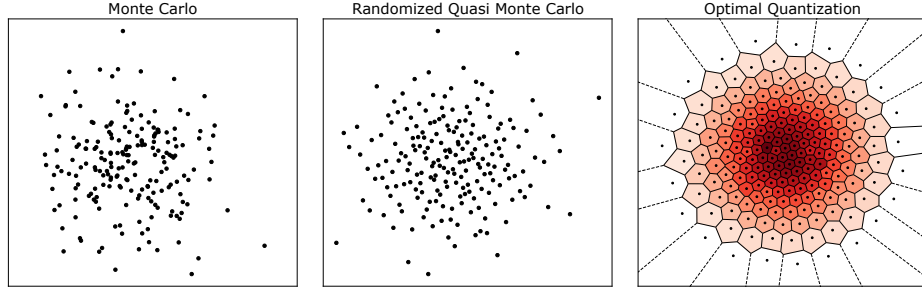

Figure 1: Monte Carlo (left), Randomized Monte Carlo (center) and Optimal Quantization with the associated Voronoi Cells (right), for a sampling size $N = 200$ of the bivariate normal distribution $\mathcal{N}(0, I_2)$.

computational libraries, Variational Inference has become one of the most prominent framework for probabilistic inference approximation.

Most of these optimization procedures rely on gradient descent optimization over the parameters associated with the variational family and subsequently depending heavily on the $\ell_2(\mathbb{R}^K)$ (with $K$ the number of variational parameters) norm of the expected gradient [5, 8]. The bias-variance decomposition gives

$$\mathbb{E}|g|_{\ell_2}^2 = \operatorname{tr}\mathbb{V}g + |\mathbb{E}g|_{\ell_2}^2. \tag{2}$$

Low variance of the gradient estimators allows for taking larger steps in the parameter space and result in faster convergence if the induced bias can be satisfyingly controlled. Several methods have been used to reduce gradient variance such as filtering [25, 33] control variate [10] or alternative sampling [37, 34, 7].

In real-world applications, one would test a large combination of models and hyperparameters associated with multiple preprocessing procedures. A common practice for bayesian modeling on large datasets consists of using VI for model selection before resorting to asymptotically exact sampling methods. More generally, VI is typically the first step towards more complex and demanding sampling. In this work we propose to give more importance to parsimonious computation than accuracy. Our approach is based upon embracing the fundamental bias in resorting to VI approach and finding the best variance free estimator which produces the fastest gradient descent. This work proposes to use Optimal Quantization (OQ) (also called Optimal Voronoi Tesselation, see [15] for an historical view) in place for the variational distribution. Given a finite subset $\Gamma_N$ of $\mathbb{R}^d$, the optimal quantizer at level $N$ of a random variable $Z \in L_{\mathbb{R}^d}^p(\Omega, \mathcal{A}, \mathbb{P})$ on a probability space $(\Omega, \mathcal{A}, \mathbb{P})$ is defined as the closest finite probability measure on $\Gamma_N$ for the $L_{\mathbb{R}^d}^p(\Omega, \mathcal{A}, \mathbb{P})$ distance. Hence, it is by construction the best finite approximation of size $N$ in the $L_{\mathbb{R}^d}^p$ sense. Recent works have shown that, given a regularity term $\alpha$, the Absolute Error error induced by such quantization is in $\mathcal{O}(N^{-\frac{1+\alpha}{d}})$ [23, 29].

**Contribution.** We show that: **i)** thanks to invariance under translation and scaling our method can be applied to a large class of variational family at similar computational cost; **ii)** even though biased our estimation is lower than the true lower bound under some assumptions with know theoretical bounds, making it relevant for quick evaluation of model; **iii)** our approach leads to competitive bias-variance trade.

**Organisation of the paper.** Section 2 introduces the idea of using Optimal Quantization for VI and shows how it can be considered as the optimal choice among variance free gradients. Section 3 is devoted to the practical evaluation of these methods and show their benefits and limitations. Due to space restrictions, all theoretical proofs and derivations are in the supplementary materials.

---

**Algorithm 1:** Monte Carlo Variational Inference.

---
**Input:** $y$, $p(x,z)$, $q_{\lambda_0}$.
**Result:** Optimal VI parameters $\lambda^*$.

**1 while** *not converged* **do**
**2** $\quad$ Sample $(X_1^{\lambda_k}, \ldots, X_N^{\lambda_k}) \sim q_{\lambda_k}$ ;
**3** $\quad$ Compute $\widehat{g}_{MC}^N(\lambda_k) = \frac{1}{N} \sum_{i=1}^{N} \nabla_\lambda H(X^{\lambda_k})$ ;
**4** $\quad$ $\lambda_{k+1} = \lambda_k - \alpha_k \widehat{g}_{MC}^N(\lambda_k)$ ;
**5 end**

---

## 2 Quantized Variational inference

In this Section, we present Quantized Variational Inference. We review traditional Monte Carlo Variational Inference in 2.1. Details of our algorithm along with theoretical results are presented in 2.2. Finally, section 2.3 proposes an implementation of Richardson extrapolation to reduce the produced bias.

### 2.1 Variational inference

Given a parameter family $\lambda \in \mathbb{R}^K$, exact estimation of equation 1 is possible in the conjugate distribution case given some models when closed-forms are avalaible [4, 38]. Complex or black box models require the use of minimum-search strategy such as Stochastic Gradient Descent (SGD), provided that a suitable form for the gradient can be found. Expressing $z$ as a transformation over a random variable $X \sim q$, which holds all the stochasticity of $z$, such as $z = h_\lambda(X)$ allows for derivation under the expectation. In this case, the gradient can be expressed as

$$\nabla_\lambda \mathscr{L}(\lambda) = \mathbb{E}_{X \sim q}[\nabla_\lambda \underbrace{(\log p(y, h_\lambda(X)) - \log q(h_\lambda(X))|\lambda)}_{H(X,\lambda)}], \tag{3}$$

clearing the way for optimization step since one only needs to compute the gradient for a batch of samples and take the empirical expectation. This is known as the reparametrization trick [21] . In the following, $H(X^\lambda)$ denotes the stochastic function $H(X, \lambda)$ with $\mathscr{L}(\lambda) = \mathbb{E}[H(X^\lambda)]$ when there is no ambiguity and $g_\lambda(X) = \nabla H(X^\lambda)$ the stochastic gradient for the ELBO maximization problem.

A typical MC procedure at step $k$ samples from an **i.i.d.** sequence $(X_1^{\lambda_k}, \ldots, X_N^{\lambda_k}) \sim q_{\lambda_k}$ and computes $\widehat{\mathscr{L}}_{MC}^N(\lambda_k) = \frac{1}{N} \sum_i^N H(X_i^{\lambda_k})$ along with $\widehat{g}_{MC}^N(\lambda_k) = \frac{1}{N} \sum_i^N \nabla H(X_i^{\lambda_k})$. Then, SGD scheme described in algorithm 1 can be used.

The convergence of the procedure typically depends on the expectation of the quadric norm of $\mathbb{E}[\widehat{g}^N]$ [18, 8]. Equation 2 shows that this method results in an MSE error of $\mathcal{O}(N^{-1})$ (by the Law Of Large Number) as the estimator is unbiased. Various methods have already been proposed to improve on this rate [25, 33, 10, 37, 34] .

Our work considers the class of variance-free estimator and aims to find the best candidate to improve on this bound, at the cost of introducing a systematic bias in the evaluation which can be reduced using Richardson extrapolation (see section 2.2).

### 2.2 Optimal Quantization

In this section we consider the true ELBO $\mathscr{L}(\lambda) = \mathbb{E}[H(X^\lambda)]$ and construct an optimal quantizer $X^{\Gamma_N,\lambda}$ of $X^\lambda$ along with an ELBO estimator $\widehat{\mathscr{L}}_{OQ}^N(\lambda) = \mathbb{E}[H(X^{\Gamma_N,\lambda})]$, such as $\|X^\lambda - X^{\Gamma_N,\lambda}\|_2$ is minimized.

**Definition 1.** *Let* $\Gamma_N = \{x_1, \ldots, x_N\} \subset \mathbb{R}^d$ *be a subset of size* $N$, $(C_i(\Gamma))_{i=1,\ldots,N} \subset \mathscr{P}(\mathbb{R}^d)$ *and*

$$\forall i \in \{1, \ldots, N\} \quad C_i(\Gamma) \subset \left\{ \xi \in \mathbb{R}^d, |\xi - x_i| \leq \min_{j \neq i} |\xi - x_j| \right\}, \tag{4}$$

*then* $(C_i(\Gamma))_{i=1,\ldots,N}$ *is a Voronoi partition of* $\mathbb{R}^d$ *associated with the Voronoi Cells* $C_i$.

Let $(\Omega, \mathscr{A}, \mathbb{P})$ be the probability space. For $X^\lambda \in L_{\mathbb{R}^d}^2(\Omega, \mathscr{A}, \mathbb{P})$, Optimal Quantization aims to find the best $\Gamma \subset \mathbb{R}^d$ of cardinality at most $N$ in $L_{\mathbb{R}^d}^2$. To that end, the optimal quantizer of $X^\lambda$ is

defined as the projection onto the closest Voronoi cell induced by $\Gamma_N$. Formally, if we consider the projection $\Pi : \mathbb{R}^d \to \mathbb{R}^d$ such as $\Pi(x) = \sum_{i=1}^N x_i \mathbb{1}(x)_{C_i(\Gamma)}$, then

$$X^{\Gamma_N, \lambda} = \Pi(X^\lambda). \tag{5}$$

The quantizer $\Gamma_N^* = (x_1, \ldots, x_N)$ of $X^\lambda$ at level $N$ is quadratically optimal if it minimizes the quadratic error $\|X^\lambda - X^{\Gamma_N, \lambda}\|_2 = \mathbb{E}\left[\min_{1 \le i \le N} |X^\lambda - x_i|^2\right]$. The problem can be reformulated as finding the probability measure on the convex subset of probability measure on $\Gamma_N$ that minimizes the $L^2_{\mathbb{R}^d}(\Omega, \mathscr{A}, \mathbb{P})$ Wassertein distance [24] .

For illustration, different sampling methods for the bivariate normal distribution $\mathcal{N}(0, I_2)$ are represented in Figure 1. It is shown that Randomized Quasi Monte Carlo produces more concentrated samples in the high density regions where Optimal Quantization accurately represents the probability distribution. Given a sample from OQ, the associated weights $\mathbb{P}\left(X^{\Gamma_N, \lambda} = x_i\right)$ gives his relative importance (values are displayed in shades of red).

Given $N$ and $\Gamma_N^*$, the error rate of such approximation is controlled by Zador's Theorem [29, 30, 27]

$$\left\| X^\lambda - X^{\Gamma_N^*, \lambda} \right\|_2 \le \mathcal{O}(N^{-\frac{1}{d}}). \tag{6}$$

The key property of the optimal quantizer lays in the simplicity of his cubature formula. For every measurable function $f$ such as $f(X) \in L^2_{\mathbb{R}^d}(\Omega, \mathscr{A}, \mathbb{P})$

$$\mathbb{E}\left[f(X^{\Gamma_N, \lambda})\right] = \sum_{i=1}^N \mathbb{P}\left(X^{\Gamma_N, \lambda} = x_i\right) f(x_i). \tag{7}$$

This result opens the possibility for using Optimal Quantizer expectation $\mathbb{E}\left[f(X^{\Gamma_N, \lambda})\right]$ as an approximation for the true expectation. As a deterministic characterization of $X^\lambda$, equation 7 can be compared to its counterpart when one considers Quasi Monte Carlo (QMC) sampling with $X^\lambda_{QMC}$ obtained from evaluating a low discrepancy sequence $\{\mathbf{u}_1, \cdots, \mathbf{u}_N\}$ with the inverse cumulative function of distribution $X^\lambda$. It results in a similar curbature formula but with equal normalized weights. This method typically produces an absolute error in $\mathcal{O}(\frac{\log(N)}{N})$. By considering relevant weights on each sample, the optimal quantization improves the estimation by a factor $\log(N)$.

**Regularity.** The precision of the approximation improves with regularity hypothesis. For instance, let $\alpha \in [0, 1], \eta \ge 0$, if $F$ is continuously differentiable on $\mathbb{R}^d$ with $\alpha$-Hölder gradient and $X \in L^{2+\eta}_{\mathbb{R}^d}(\mathbb{P})$, one has the following bound on the Absolute Error [27]

$$\left|\mathbb{E}F(X) - \mathbb{E}F\left(\widehat{X}_N^\Gamma\right)\right| \le C_{d,\mu} [\nabla F]_\alpha N^{-\frac{1+\alpha}{d}}. \tag{8}$$

**Getting Optimal Quantization.** The main drawback of Optimal Quantization is the computational cost associated with constructing an optimal N-quantizer $X^{\Gamma_N, \lambda}$ compared to sampling from $X^\lambda$. Even though it is time-consuming in higher dimensions, one must keep in mind that it can be built offline and that efficient methods exist to approximate the optimal quantizer. For instance, K-means are used to obtain such grid at a reasonable cost of $\mathcal{O}(N \log N)$ [13]. Moreover, in the context of AVI with normal approximation, it is possible to rely solely on $D$ dimensional normal grid to perform optimization since every normal distribution can be obtained by shifting and scaling. The same goes for every distribution that can be determined by such transformation of a base random variable. Note that the optimal grid for the normal distribution can be downloaded for dimensions up to 10 (`http://www.quantize.maths-fi.com/downloads`).

**Quantized Variational Inference.** The curbature formula 7 is used to compute the OQ expectation at a similar cost than regular MC estimation. Replacing the MC term in equation 3 by its quantized counterpart is straightforward. The quantized ELBO estimator is defined by

$$\widehat{\mathscr{L}}_{OQ}^N(\lambda) = \sum_{i=1}^N \omega_i H\left(X_i^{\Gamma, \lambda}\right). \tag{9}$$

A crucial point is that the quantized ELBO is always lower than the expected one under the assumption of convex ELBO objective. This particular point justifies the usefullness of the method for quick evalutation of model performance.

---

**Algorithm 2:** Quantized Variational Inference.

---
**Input:** $y$, $p(x,z)$, $q_{\lambda_0}$.
**Result:** Optimal Quantized VI parameters $\lambda_q^*$.

---
1 **while** *not converged* **do**
2     Get $(X_1^{\Gamma_N,\lambda_k}, \ldots, X_N^{\Gamma_N,\lambda_k}) \sim q_{\lambda_k}$, $(w_1^k, \ldots, w_N^k)$ ;
3     Compute $\widehat{g}_{OQ}^N(\lambda_k) = \nabla_\lambda \sum_{i=1}^N w_i^k H(X_i^{\Gamma_N,\lambda_k})$ ;
4     $\lambda_{k+1} = \lambda_k - \alpha_k \widehat{g}_{OQ}^N(\lambda_k)$ ;
5 **end**

---

**Proposition 1.** *Let* $X^\lambda \in L_{\mathbb{R}^d}^2(\Omega, \mathcal{A}, \mathbb{P})$ *and* $X^{\Gamma_N,\lambda}$ *the associated optimal quantizer, under the hypothesis that H (Eq. 3) is a convex lipschitz function,*

$$\widehat{\mathscr{L}}_{OQ}^N(\lambda) \le \mathscr{L}(\lambda). \tag{10}$$

In fact, for proposition 1 to be true $X^{\Gamma_N,\lambda}$ needs only to fulfill the stationnary property which is defined by $\mathbb{E}\left[X^\lambda | X^{\Gamma_N,\lambda}\right] = X^{\Gamma_N,\lambda}$. Intuitively, the stationnary condition expresses the fact that the quantizer $X^{\Gamma_N,\lambda}$ is the expected value under the subset of events $\mathscr{C} \in \mathcal{A}$ such as $\Pi(X^\lambda) = X^{\Gamma_N,\lambda}$. It can be shown that the optimal quantizer has this property [14, 26] .
Computing the gradient in the same fashion leads to algorithm 2. An immediate consequence of proposition 1 is that for $\lambda_q^*$ the optimal parameters estimated from algorithm 2 and $\lambda^*$ the true optimum, we can state the following proposition

**Proposition 2.** *Let* $\lambda^* = \max_{\lambda \in \mathbb{R}^K} \mathscr{L}(\lambda)$ *and* $\lambda_q^* = \max_{\lambda \in \mathbb{R}^K} \widehat{\mathscr{L}}_{OQ}^N(\lambda)$. *Under the same assumptions than proposition 1,*

$$\mathscr{L}(\lambda^*) - \widehat{\mathscr{L}}_{OQ}^N(\lambda_q^*) \le C \left[ 2\|X^{\lambda^*} - X^{\Gamma,\lambda^*}\|_2 + \|X^{\lambda_q^*} - X^{\Gamma,\lambda_q^*}\|_2 \right]. \tag{11}$$

The approximation error of the resulting estimation follows from the Zador theorem (Eq. 8) and is in $\mathcal{O}(N^{-\frac{2(1+\alpha)}{d}})$ in term of MSE depending on the regularity of $H$. The crucial implication of proposition 2 is that relative model performance can be evaluated with our method. Poor relative true performance, provided that the difference in terms of ELBO minimum sufficiently large in regard of the approximation error, produces poor relative performance with Quantized Variational Inference.

Performing algorithm 2 implies finding the new optimal quantizer for $X^{\Gamma,\lambda_k}$ at each step $k$. We highlight that the competitiveness of the method in terms of computational time is due to the fact that optimal quantizer derived from the base distribution $\mathbb{P}_X$ can be used to obtain $X^{\Gamma,\lambda}$ when $X^\lambda$ can be obtained through scaling and shifting of $X$, since optimal quantization is preserved under these operations. For instance, in the case of BBVI with Gaussian distribution, we only need the optimal grid $X^\Gamma$ of $\mathcal{N}(0, I_d)$ and use $X^{\Gamma,\lambda} = \mu + X^\Gamma \Sigma^{\frac{1}{2}}$ (given $\Sigma^{\frac{1}{2}}$ the Cholesky decomposition of $\Sigma$) to obtain the new optimal quantizer. The same goes for the distributions in the exponential family. Details about the optimal quantization for the gaussian case can be found in [30]. Thus, this method applies to a large class of commonly used variational distributions.

The previous results imply that quantization is relevant only for $d < 2(1+\alpha)$ compared to MC sampling. However, numerous empirical studies have shown that this bound may be overly pessimistic, even for a not so sparse class of function in $L_{\mathbb{R}^d}^2$ [29]. Going further, we can implement Richardson extrapolation to improve on this bound.

## 2.3 Richardson Extrapolation

Richardson extrapolation [32] was originally used for improving the precision of numerical integration. The extension to optimal quantization was first introduced in [29, 28] in the finance area to bring an answer to expensive computation of some expectation $\mathbb{E}\left[f(X_T)\right]$ for a diffusion process $X_t$ representing a basket of assets and $f$ an option with maturity $T$.

Richardson extrapolation leverages the stationary property of an optimal quantizer through error expansion. We illustrate in the one-dimensional case. Let $H$ be twice differential function with lipschiptz continuous second derivative. By Taylor's expansion

$$\mathbb{E}\left[H(X^\lambda)\right] = \mathbb{E}\left[H(X^{\Gamma_N,\lambda})\right] + \mathbb{E}\left[H'(X^{\Gamma_N,\lambda})(X^\lambda - X^{\Gamma_N,\lambda})\right]$$
$$+ \mathbb{E}\left[H''(X^{\Gamma_N,\lambda})(X^\lambda - X^{\Gamma_N,\lambda})^2\right] + \mathcal{O}(\mathbb{E}\left[|X^\lambda - X^{\Gamma_N,\lambda}|^3\right]).$$

Then, using the stationnary property, the first order term vanishes since

$$\mathbb{E}\left[(X^\lambda - X^{\Gamma_N,\lambda})\right] = \mathbb{E}\left[\mathbb{E}\left[(X^\lambda - X^{\Gamma_N,\lambda})|X^{\Gamma_N,\lambda}\right]\right]$$
$$= \mathbb{E}\left[\mathbb{E}\left[X^\lambda|X^{\Gamma_N,\lambda}\right] - X^{\Gamma_N,\lambda}\right]$$
$$= 0.$$

Taking two optimal quantizer $X^{\Gamma_N,\lambda}$ and $X^{\Gamma_M,\lambda}$ of $X^\lambda$ at level $N, M$ with $N \geq M$ and using the fact that $\mathbb{E}\left[|X^\lambda - X^{\Gamma_N,\lambda}|^3\right] = \mathcal{O}(N^{-3})$ [16], it is possible to eliminate the first order term by combining the two estimators with a factor $N^2$ and $M^2$.

$$\mathscr{L}(\lambda) = \frac{N^2 \widehat{\mathscr{L}}_{OQ}^N(\lambda) - M^2 \widehat{\mathscr{L}}_{OQ}^M(\lambda)}{N^2 - M^2} + \mathcal{O}(N^{-1}\left(N^2 - M^2\right)^{-1}). \tag{12}$$

We generally take $\frac{N}{M} = \gamma$ with $\gamma \in [1,2]$ due to additional computational cost. For instance, taking $N = 2M$ leads to $\mathcal{O}(N^{-3})$ in term of absolute error. Recent results [29, 23] in higher dimension show that the general error is $\mathcal{O}(N^{-\frac{2}{d}}(N^{\frac{2}{d}} - M^{\frac{2}{d}})^{-1})$. Even though $\gamma = 2$ led to satisfying results in our experiments, applying this method to VI can lead to computational instability in higher dimensions and there is no straightforward method for finding the optimal $\gamma$.

## 3   Experiments

To demonstrate the validity and effectiveness of our approach, we considered Bayesian Linear Regression (BLR) on various dataset, a Poisson Generalized Linear Model (GLM) on the frisk data and a Bayesian Neural Network (BNN) on the metro dataset. For $q_\lambda$, we choose the standard Mean-Field variational approximation with Gaussian distributions.

**Setup.**   Experiments are performed using python 3.8 with the computational library Tensorflow [1]. Adam [19] optimizer is used with various learning rates $\alpha$ and default $\beta_1 = 0.9, \beta_2 = 0.999$ values recommended by the author. The benchmark algorithms comprises the traditionnal MCVI described in algorithm 1, RQMC considered in [7] and QMC. We underline that [7] shows that RQMC outperforms state of the art control variate techniques such as Hessian Vector Product (HPV) [25] in a similar setting. We compare it with the implementation of algorithm 2 (QVI) and the Richardson extrapolation RQVI. For all experiments we take a sample size $N = 20$. When $D \leq 10$, precomputed optimal quantizer available online [1] is used. The Optimal Quantization is approximated in higher dimension using the R package muHVT. The number of parameters $K$ along with the number of samples for each dataset is reported in Table 1. The complete documented source code to reproduce all experiments is available on GitHub [2].

Table 1: Datasets used for the experiments along with the Relative Bias (RB) at the end of execution for QVI and RQVI using the best learning rate.

| Dataset | Size | K | QVI RB | RQVI RB |
|---|---|---|---|---|
| Boston | 506 | 18 | **13%** | 7% |
| Fires | 517 | 16 | **3%** | 1% |
| Life Expect. | 2938 | 36 | **0.3%** | 0.04% |
| Frisk | 96 | 70 | **6%** | |
| Metro | 48204 | 60 | **5%** | |

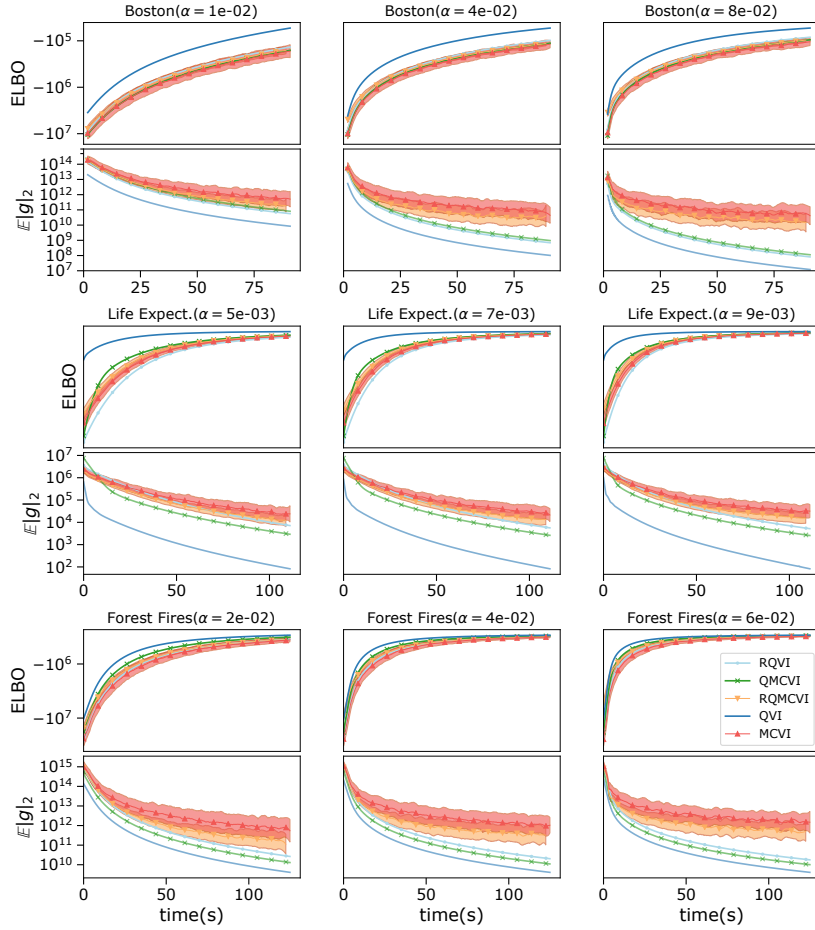

Figure 2: **Bayesian Linear Regression**. Evolution of the ELBO (odd rows, log scale) and expect gradient norm (even rows, log scale) during the optimization procedure for datasets reported in Table 1 using Adam for MCVI (red), RQMCVI (orange), QMCVI (green), QVI (blue), RQVI (light blue) as function of time. Variance for MC estimator (red area) and RQMC (orange area) are obtained by 20 runs of each experiment.

**Bayesian Linear Regression.**    Figure 2 shows the evolution of the ELBO along with the expected $\ell_2$ norm of the gradient $\mathbb{E}|g|^2_{\ell_2}$, both in log-scale. We see that QVI converges faster than vanilla MCVI and the baseline on all datasets. The gradient of both QVI and RQVI is lower than MCVI thanks to the absence of variance. However, only QVI performs better than MCVI on all datasets. For all learning rates $\alpha$ considered, the expected norm of the gradient is significantly lower. In these examples, it appears that the gain obtained from using RQVI is lost in the additional computation required for this method. We observe that using RQMC sampling reduces the gradient variance (odd rows) and improves the convergence rate for all experiments.

In these experiments, the resulting bias after performing a complete Gradient Descent is relatively small compared to the starting value of the ELBO. The resulting biases are reported in Table 1 and span from almost 0 for the Life Expectancy dataset to 13% for the Boston dataset. The fact that $\widehat{\mathscr{L}}^N_{OQ}(\lambda) > \mathscr{L}(\lambda)$ is a consequence of proposition 2.

**Poisson Generalized Linear Model.**    Similar results are obtained by QVI for the GLM model on Frisk dataset (see Figure 3). QMCVI perform similarly to QVI for all learning rates but produces a larger bias in the ELBO objective function. As mentionned, RQVI can be computationnaly instable as the dimension grows. Indeed, denoting $\gamma = \frac{N}{M}$ and $\epsilon = \frac{2}{D}$, computing ELBO with Richardson

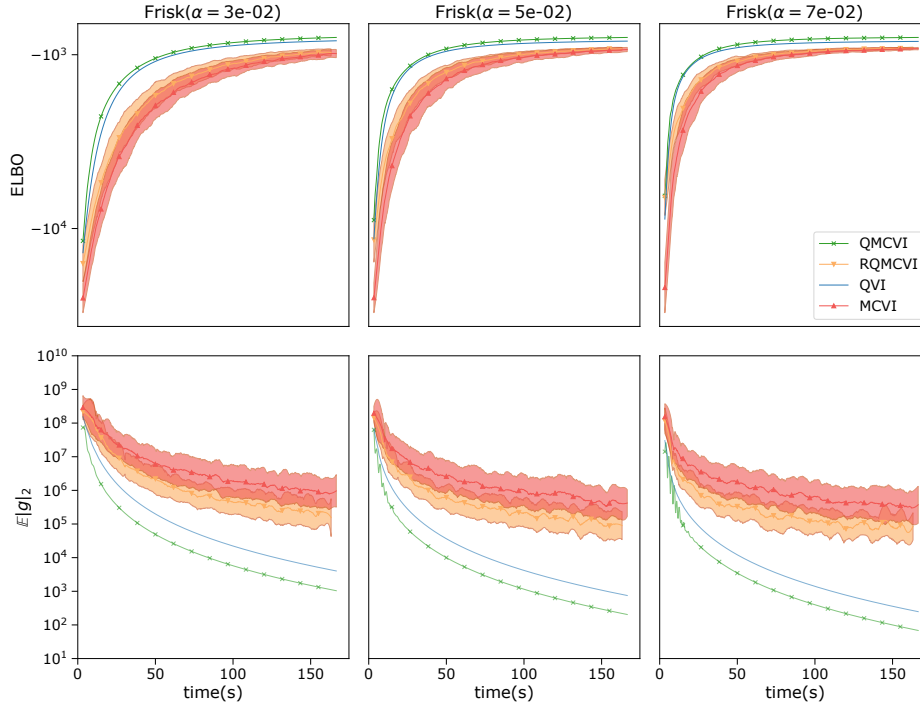

Figure 3: **Generalized Poisson Regression**. Evolution of the ELBO in (first row, in log scale) and expect gradient norm (second row, in log scale) during the optimization procedure for the frisk datasets (see Table 1) using Adam for MCVI (red), RQMCVI (orange), QMCVI (green), QVI (blue) as function of time. QVI exhibits comparable performance to QMCVI for all selected learning rate $\alpha$. We use $N = 20$ sample for each experiments. Using QVI produces a relative bias of 6%.

extrapolation leads to

$$\widehat{\mathscr{L}}(\lambda) = \frac{\gamma^\epsilon \widehat{\mathscr{L}}^N_{OQ}(\lambda) - \widehat{\mathscr{L}}^M_{OQ}(\lambda)}{\gamma^\epsilon - 1}. \tag{13}$$

For large $D$, even a small computational error between $\widehat{\mathscr{L}}^N_{OQ}(\lambda)$ and $\widehat{\mathscr{L}}^M_{OQ}(\lambda)$ can produce a large error in the estimation of $\widehat{\mathscr{L}}(\lambda)$ which led to the failure of the procedure.

**Bayesian Neural Network.** Finally, Bayesian Neural Network model is tested against the baseline. It consists of a Multi Layer Perceptron composed of 30 ReLu activated neurons with normal prior on weights and Gamma hyperpriors on means and variances. Inference is performed on the metro dataset. Similarly to the other experiments, Figure 4 shows that QVI converges faster than the baseline for all hyperparameters considered in only few epochs. Quantitatively, by taking $\alpha = 7e-3$ we can see that a stopping rule on the evolution of the parameters $\lambda_k$, the gradient descent procedure would terminate at $t \approx 100$ seconds for QVI and $t \approx 500$ (seconds) for the MCVI algorithm.

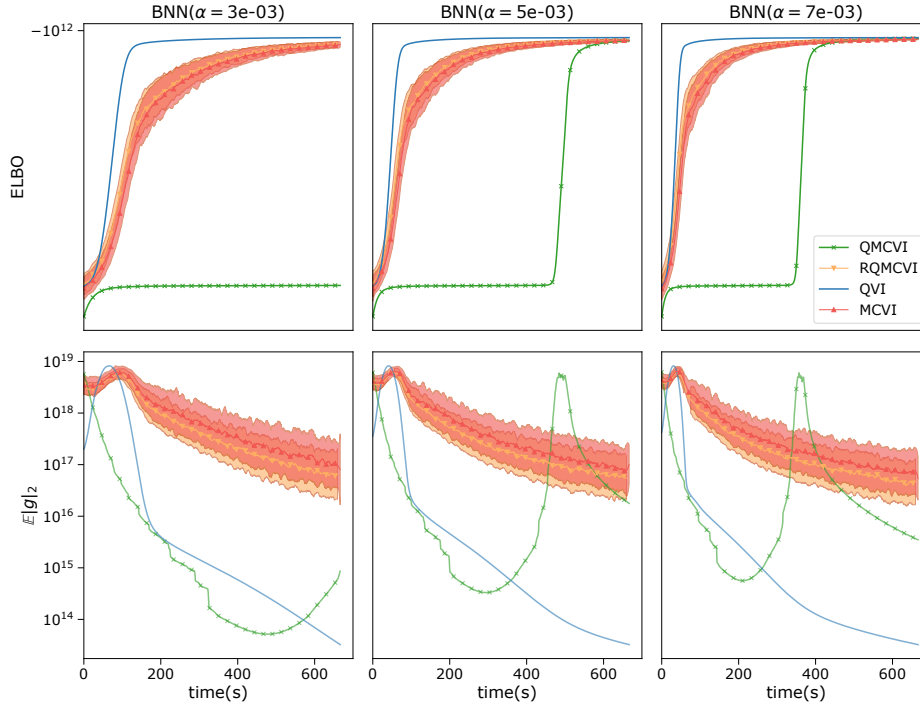

Figure 4: **Bayesian Neural Network**. Evolution of the ELBO in (first row, in log scale) and expect gradient norm (second row, in log scale) during the optimization procedure for the metro datasets (see Table 1) using Adam for MCVI (red), RQMCVI (orange), QMCVI (green), QVI (blue) as function of time. QVI exhibits superior performance with all selected learning rate $\alpha$. We use $N = 20$ sample for each experiments. Using QVI produces a relative bias of 10%.

## 4  Conclusion

This work focuses on obtaining a variance-free estimator for the ELBO maximization problem. To that end, we investigate the use of Optimal Quantization and show that it can lead to faster convergence. Moreover, we provide a theoretical guarantee on the bias and regarding its use as an evaluation tool for model selection.

The base QVI algorithm can be implemented with little effort in traditional VI optimization package as one only needs to replace MC estimation with a weighted sum.

Various extensions could be proposed, including a simple quantized control variate using the optimal quantized to reduce variance or Multi-step Richardson extrapolation [9]. In addition, this method could be applied more broadly to any optimization scheme, where sampling has a central role, such as normalizing flow or Variational Autoencoder. We plan to consider it in future work.

## 5 Broader Impact

Our work provides a method to speed up the convergence of any procedure involving the computation of an expectation on a large distribution class. Such case corresponds to a broad range of applications from probabilistic inference to pricing of financial products [29]. More generally, we hope to introduce the concept of optimal quantizer to the machine learning community and to convince of the value of deterministic sampling in stochastic optimization procedures.

Reducing the computational cost associated with probabilistic inference allows considering a broader range of models and hyperparameters. Improving goodness of fit is the primary goal of any statistician and virtually impacts all aspects of social life where such domain is applied. For instance, we chose to consider the sensitive subject of the New York City Frisk and Search policy in the 1990s. In-depth analysis of the results shows that minority groups are excessively targeted by such measure even after controlling for precinct demographic and ethnic-specific crime participation [11]. This study gave a strong statistical argument to be presented to the authorities for them to justify and amend their policies.
Even though environmental benefits could be argued, we do not believe that such benefits can be obtained through increased efficiency of a system due to the rebound effect.

In the paper, we stressed the benefit of using our approach to improve automated machine learning pipelines, which consider large classes of models to find the best fit. This process can remove the practitioner from the modeling process, overlook any ML model's inherent biases, and ignore possible critical errors in the prediction. We strongly encourage practitioners to follow standard practices such as posterior predictive analysis and carefully examine the chosen model's underlying hypothesis.

## 6 Acknowledgments

The author thanks Mathilde Mougeot for the invaluable insights and corrections. The author is grateful to the reviewers whose particularly relevant comments improved this work. This research was supported by the French National Railway Company (SNCF) and has been partially funded by the «Industrial Data Analytics and Machine Learning» Chair of ENS Paris-Saclay.

## Footnotes

[1] http://www.quantize.maths-fi.com/downloads

[2] https://github.com/amirdib/quantized-variational-inference

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
