[Supplementary Material 1]

# Supplementary Material of Quantized Variational Inference

## A   ELBO derivation

Assumes that we have observations $y$, latent variables $z$ and a model $p(y,z)$ with $p$ the density fonction for the distribution $y$. By Bayes' Theorem

$$p(z|y) = \frac{p(y|z)p(z)}{p(y)}$$
$$= \frac{p(y|z)p(z)}{\int_z p(z,y)\,dz}.$$

Using the definition of KL divergence,

$$\begin{aligned}
\mathrm{KL}[q_\lambda(z)\|p(z|y)] &= \int_z q_\lambda(z)\log\frac{q_\lambda(z)}{p(z|y)}\,dz \\
&= -\int_z q_\lambda(z)\log\frac{p(z|y)}{q_\lambda(z)}\,dz \\
&= -\int_z q_\lambda(z)\log\frac{p(z,y)}{q_\lambda(z)}\,dz + \int_z q_\lambda(z)\log p(y)\,dz \\
&= -\int_z q_\lambda(z)\log\frac{p(z,y)}{q_\lambda(z)}\,dz + \log p(y)\int_z q_\lambda(z)\,dz \\
&= -\mathscr{L}(\lambda) + \log p(y).
\end{aligned}$$

Rearranging the terms gives equation (1).

## B   Proofs

Let $f(X) \in L^2_{\mathbb{R}^d}(\Omega, \mathscr{A}, \mathbb{P})$ and $X^{\Gamma_N,\lambda}$ the the optimal quantizer of $X^\lambda$. The general framework of our study can be stated as estimating the quantity

$$I = \mathbb{E}\left[f(X)\right]. \tag{1}$$

We define the *MC* and *OQ* estimators as

$$I_{MC} = \frac{1}{N}\sum_{i=1}^{N} f(X_i), \tag{2}$$

$$I_{OQ} = \sum_{i=1}^{N} \underbrace{\mathbb{P}\left(X^{\Gamma_N,\lambda} = x_i\right)}_{\omega_i} f(x_i). \tag{3}$$

It is direct to derive $\|I - I_{MC}\|_2 = \mathcal{O}(N^{-\frac{1}{2}})$. In the following we establish the approximation error for the $I_{OQ}$ estimator.

In this part we demonstrates proposition 1 and proposition 2. The former is particularly important since it establishes an asymptomatic bound on the error produced by using QVI. When considering

it along with proposition 1 justifies QVI, for ranking models with it will produce true ranking provided that the relative difference in ELBO is lower than the quantization error. In the following we formally demonstrate such result (thorough investigation of optimal quantizer can be found in [8, 7]). We begin with the definition of a stationnary quantizer.

**Definition 1.** *Let* $\Gamma_N = \{x_1, \ldots, x_N\}$ *be a quantization scheme of* $X^\lambda$. $X^{\Gamma_N, \lambda}$ *is said to be stationary quantizer if the Voronoi partition induced by* $\Gamma_N$ *satisfies* $\mathbb{P}(X \in C_i(x)) > 0 \ \forall i \in \{1, \ldots, N\}$ *and*

$$\mathbb{E}\left[X^\lambda | X^{\Gamma_N, \lambda}\right] = X^{\Gamma_N, \lambda}.$$

One of the first question raised by using optimal quantization $\mathbb{E}\left[H(X^{\Gamma_N, \lambda})\right]$ in place for $\mathbb{E}\left[H(X^\lambda)\right]$ is the error produced by such substitution. Let us remind that we denote $\widehat{\mathscr{L}}_{OQ}^N(\lambda) = \mathbb{E}\left[H(X^{\Gamma_N, \lambda})\right]$ the quantized ELBO estimator and $\mathscr{L}(\lambda) = \mathbb{E}\left[H(X^\lambda)\right]$ the true ELBO.

**Lemma 1.** *Let* $X^\lambda \in L_{\mathbb{R}^d}^2(\Omega, \mathscr{A}, \mathbb{P})$ *and a* $H$ *a continuous lipschitz function with Lipschitz constant* $C$, *we have*

$$\left|\mathscr{L}(\lambda) - \widehat{\mathscr{L}}_{OQ}^N(\lambda)\right| \leq C \left\|X^\lambda - X^{\Gamma_N, \lambda}\right\|_2.$$

*Proof.*

$$\left|\mathbb{E}\left[H(X^\lambda)\right] - \mathbb{E}\left[H(X^{\Gamma_N, \lambda})\right]\right| \leq \mathbb{E}\left[\mathbb{E}\left[\left|H(X^\lambda) - H(X^{\Gamma_N, \lambda})\right| | X^{\Gamma_N, \lambda}\right]\right] \qquad (4)$$

$$\leq C \left\|X^\lambda - X^{\Gamma_N, \lambda}\right\|_1$$

$$\leq C \left\|X^\lambda - X^{\Gamma_N, \lambda}\right\|_2. \qquad (5)$$

We use Jensen inequality in equation 4 and the monoticity of the $L_p(\Omega, \mathscr{A}, \mathbb{P})$ norm as a function of $p$ in equation 5. $\square$

**Proposition 1.** *Let* $X^\lambda \in L_{\mathbb{R}^d}^2(\Omega, \mathscr{A}, \mathbb{P})$ *and* $X^{\Gamma_N, \lambda}$ *the associated optimal quantizer, under the hypothesis that* $H$ *is a convex lipschitz function,*

$$\widehat{\mathscr{L}}_{OQ}^N(\lambda) \leq \mathscr{L}(\lambda).$$

*Proof.*

$$\widehat{\mathscr{L}}_{OQ}^N(\lambda) = \mathbb{E}\left[H(X^{\Gamma_N, \lambda})\right]$$

$$= \mathbb{E}\left[H\left(\mathbb{E}\left[X^\lambda | X^{\Gamma_N, \lambda}\right]\right)\right] \qquad (6)$$

$$\leq \mathbb{E}\left[\mathbb{E}\left[H(X^\lambda) | X^{\Gamma_N, \lambda}\right]\right]$$

$$= \mathbb{E}\left[H(X^\lambda)\right] \qquad (7)$$

$$= \mathscr{L}(\lambda)$$

When we used Lemma 1 in equation 6 and the conditional Jensen inequality to obtain 7. $\square$

**Proposition 2.** *Let* $\lambda^* = \min_{\lambda \in \mathbb{R}^K} \mathscr{L}(\lambda)$ *and* $\lambda_q^* = \min_{\lambda \in \mathbb{R}^K} \widehat{\mathscr{L}}_{OQ}^N(\lambda)$. *Under the same assumptions than proposition 1,*

$$\mathscr{L}(\lambda^*) - \widehat{\mathscr{L}}_{OQ}^N(\lambda_q^*) \leq C\left[2\|X^{\lambda^*} - X^{\Gamma, \lambda^*}\|_2 + \|X^{\lambda_q^*} - X^{\Gamma, \lambda_q^*}\|_2\right].$$

*Proof.* A immediate consequence of proposition 1 is that $\widehat{\mathscr{L}}_{OQ}^{N}(\lambda_q^*) \leq \mathscr{L}(\lambda^*)$. Then, we can write

$$
\begin{aligned}
\mathscr{L}(\lambda^*) - \widehat{\mathscr{L}}_{OQ}^{N}(\lambda_q^*) = {} & \mathscr{L}(\lambda^*) - \widehat{\mathscr{L}}_{OQ}^{N}(\lambda^*) \\
& + \widehat{\mathscr{L}}_{OQ}^{N}(\lambda^*) - \mathscr{L}(\lambda_q^*) \\
& + \mathscr{L}(\lambda_q^*) - \widehat{\mathscr{L}}_{OQ}^{N}(\lambda_q^*) \\
\leq {} & C\|X^{\lambda^*} - X^{\Gamma,\lambda^*}\|_2 \\
& + C\|X^{\lambda_q^*} - X^{\Gamma,\lambda_q^*}\|_2 \\
& + C\|X^{\lambda^*} - X^{\Gamma,\lambda^*}\|_2
\end{aligned}
$$

Using Lemma 1 and noting that

$$
\widehat{\mathscr{L}}_{OQ}^{N}(\lambda^*) - \mathscr{L}(\lambda_q^*) \leq \widehat{\mathscr{L}}_{OQ}^{N}(\lambda^*) - \mathscr{L}(\lambda^*),
$$

proposition 2 follows. □

Finally, Zador's theorem is used to derive non-asymptotic bound (see [5] for a complete proof).

**Theorem 1** (Zador's Theorem). *Let $X^\lambda \in L_{\mathbb{R}^d}^2(\Omega, \mathscr{A}, \mathbb{P})$ and $X^{\Gamma_N,\lambda}$ the associated optimal quantizer at level N, there exists a real constant $C_{d,p}$ such that*

$$
\forall N \geq 1, \quad \left\| X - \widehat{X}^{\Gamma_x} \right\|_p \leq C_{d,p} N^{-\frac{1}{d}}
$$

Where $C_{d,p}$ dependens only $d$ and $p$. This result can be vastly improved when $H$ exhibits more regularity. For instance, if H is an $\alpha$ hölderian function, we can obtain a bound in $\mathcal{O}(N^{-\frac{1+\alpha}{d}})$ [7].

## C Experiments

**Bayesian Linear Regression.** We used three different real-world dataset, namely Forests Fire, Boston housing datasets from the UCI repository [2] and Life Expectancy dataset from the Global Health Observatory repository. The generative Bayesian Linear Gaussian Model used is as follow.

$$
\begin{aligned}
\mathbf{b}_i &\sim \mathcal{N}\left(\mu_\beta, \sigma_\beta\right), && \text{intercepts} \\
y_i &\sim \mathcal{N}\left(\mathbf{x}_i^\top \mathbf{b}_i, \epsilon\right), && \text{output}
\end{aligned}
$$

Let $D$ be the dimension of the feature space. The dimension of the parameter space for a gaussian variationnal distribution under the mean-field assumption is $K = 2D$.

**Poisson Generalized Linear Model.** The frisk dataset is a record of stops and searches practice on civilians in New York City for fifteen months in $1998 - 1999$. It contains information about locations, ethnicity and crime statistics for each area. The question is whether these stops targeted particular groups after taking into account population and crime rates in each group for a particular precinct.
We can trace back the use of Poisson Generalized Linear Model for this use case to [3]. The model writes as follow

$$
\begin{aligned}
\mu &\sim \mathcal{N}\left(0, 10^2\right) && \text{mean offset} && (8) \\
\log\sigma_\alpha^2, \log\sigma_\beta^2 &\sim \mathcal{N}\left(0, 10^2\right) && \text{group variances} && (9) \\
\alpha_e &\sim \mathcal{N}\left(0, \sigma_\alpha^2\right) && \text{ethnicity effect} && (10) \\
\beta_p &\sim \mathcal{N}\left(0, \sigma_\beta^2\right) && \text{precinct effect} && (11) \\
\log\lambda_{ep} &= \mu + \alpha_e + \beta_p + \log N_{ep} && \text{log rate} && (12) \\
Y_{ep} &\sim \text{Poisson}\left(\lambda_{ep}\right) && \text{stops events} && (13) \\
& && && (14)
\end{aligned}
$$

$Y_{ep}$ denotes the number of frisk events for the ethnic group $e$ in the precinct $p$. $N_{ep}$ is the number of arrests for the ethnic group $e$ in the precinct $p$. Hence, in this model, $\alpha_e$ and $\alpha_p$ represents the ethnicity and precinct effect. The dataset contains three ethnicities and thirty-two precinct, which therefore exhibits $K = 70$ variational parameters.

**Bayesian Neural Network.** The Bayesian Neural Network (BNN) consists of a Multi Layer Perceptron (MLP) $\psi$ of 30 ReLU activated neurons with normal prior weights and inverse Gamma hyperprior on the mean and variance. Regression is performed on the metro dataset.

$$\alpha \sim \text{Gamma}(1, 0.1) \qquad \text{weights hyper prior} \qquad (15)$$

$$\tau \sim \text{Gamma}(1, 0.1) \qquad \text{group variances} \qquad (16)$$

$$w \sim \mathcal{N}\left(0, \frac{1}{\alpha}\right), \qquad \text{neural network weights} \qquad (17)$$

$$y \sim \mathcal{N}\left(\psi(w, x), \frac{1}{\tau}\right) \qquad \text{output} \qquad (18)$$

## D   Thanks to open source libraries

This work and many others would have been impossible without free, open-source computational frameworks and libraries. We particularly acknowledge Python 3 [9], Tensorflow [1], Numpy [6] and Matplotlib [4].

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

[Supplementary Material 2 · glr.pdf]



Boston($\alpha = 1e\text{-}02$)  Boston($\alpha = 4e\text{-}02$)  Boston($\alpha = 8e\text{-}02$)

Life Expect.($\alpha = 5e\text{-}03$)  Life Expect.($\alpha = 7e\text{-}03$)  Life Expect.($\alpha = 9e\text{-}03$)

Forest Fires($\alpha = 2e\text{-}02$)  Forest Fires($\alpha = 4e\text{-}02$)  Forest Fires($\alpha = 6e\text{-}02$)

RQVI
QMCVI
RQMCVI
QVI
MCVI