[Reviews · NeurIPS 2020]

Review 1

Summary and Contributions: This work proposes Quantized VI which uses optimal quantization for variational inference. In general, it replaces regular MC sampling in BBVI with optimal quantization. It further utilizes Richardson extrapolation to reduce bias caused by quantization. ~~~~update~~~~ I appreciate the authors' effort on the additional experiments in the rebuttal. Adding the experiments with BNN and larger datasets with more baselines indeed will make the work stronger. As the method cannot be applied out of exponential family(otherwise, it would be not affordable) as well as the batching problem pointed out by other reviewers, the method may have too limited application impact. I would increase my score from 5 to 5.5(if it allows). I appreciate the nice use of of optimize quantization in VI. However, I still cannot support the paper to be accepted at the current stage as the advantage (fast convergence) is not applicable to non-exponential models with SVI setting where it matters.

Strengths: 1. The choice of mathematical tools Optimal quantization and Richarardson expansion is suitable for the purpose that the author poses. 2. There is clear theoretical analysis of the method 3. Experimental results shows promising performance given small data setting. 4. Variational Inference is important to the community

Weaknesses: 1. Optimal quantization is not scalable (which is mentioned in the paper as well). Even with clustering before, it is costly to both N(number of data) and M(the dimension). The paper (in abstract and intro) aims to speed up VI by fast convergence which is needed for big data/big model setting, which the quantization is a bottleneck for it, which makes the method loses its point. 2. Apart form the scalability, I wonder about the effectiveness in high dimensional space as well where everything is far away from each other. 3. The experiments are only with very simple small UCI datasets and very simple/small models (linear regression). I would be great to see with more "real-life" experiments. 4. There is also limited baselines. [a] is discussed in the paper but not compared. Only the basic BBVI is compared. It would be good to see at least baselines such as [a] and [b] in the experiments. 5. For algorithm 2, it would be insanely expensive if quantization needs be to computed every round. but it is explained with exponential family, it only need once. But if it limits to be exponential family, then the point of whole BBVI is lost. 5. Small things: line 2, minimize->maximize; can you explicitly discuss about the optimal quantization computational complexity. [a]Alexander Buchholz, Florian Wenzel, and Stephan Mandt. “Quasi-Monte Carlo Variational 238 Inference”. [b] Stochastic Learning on Imbalanced Data: Determinantal Point Processes for Mini-batch Diversification

Correctness: The method seems correct.

Clarity: Clear in general.

Relation to Prior Work: Some of them are discussed but not compared.

Reproducibility: Yes

Additional Feedback:


Review 2

Summary and Contributions: The paper proposes a new optimization method that is based on sampling latent variables (they focus on the ELBO in variational inference). Their approach is based on constructing a optimal Voronoi tessellation that leads to biased but variance free gradient estimates.

Strengths: * The paper communicates the idea of Voronoi tessellation that might be new to many ML researchers.

Weaknesses: * The relevance of the setting is not clear to me (see detailed comments). Does this approach really holds what it promises -- it is not clear that really helps with "quick model checking". * The method seems to be limited to full-batch gradients, but most modern applications of VI include mini-batch sampling.

Correctness: All claims and derivations seem to be correct.

Clarity: The paper is well written and easy to follow.

Relation to Prior Work: The related work is clearly discussed.

Reproducibility: Yes

Additional Feedback: 1. Is the variance of the gradient estimator the real bottleneck for faster optimization? * Especially for reparameterization gradients I found in my experience that in most cases it has quite small variance wrt. to the latent variable sampling. So is this really an issue? * To clarify, you could plot the variance of the MCVI gradient (not only the gradient norm). 2. Application to mini-batch gradients? * Can you apply your approach also to the setting of SVI (i.e. stochastic mini-batch gradients) which is used in most cases in practice? * In this setting the mini-batch noise would probably dominate over latent variable sampling noise. Would you still see a benefit of using your method? 3. Experiments * Experiments on datasets that go beyond a couple of thousands of data points would be nice (especially, since in this regime it might be more relevant to have method for quicke model checking.) * Can you provide an experiments that really leverages your claim of quick model checking. E.g., tuning hyperparameters faster than if you would use regular MCVI? * Please quantify this gain in quicker evaluation time. AFTER REBUTTAL I thank the authors for their detailed rebuttal. I have decided to keep my initial score since I'm still not convinced of the motivation of the method/setting. To me the benefit of variance reduction (at the cost of bias) seems to be not really shown in the setting. Hower, I think the paper is an interesting read and I would agree on accepting it if that's the overall consensus.


Review 3

Summary and Contributions: The paper introduces a optimal quantization approach to estimate deterministically the gradient of the ELBO in a black-box VI setting. Edit: I updated my score to 7 as all my concerns have been properly addressed. The new experiment test the method in a challenging setting and the chosen baselines are very relevant.

Strengths: The new approach is simple and theoretically motivated. Everything can be quite straightforwardly implemented in a very generic automatic inference framework simply by pre-computing the optimal quantization for a large family of distributions. This makes the approach very suitable for probabilistic programming.

Weaknesses: The experiment section is suggestive but far too limited. No comparison with the many existing variance reduction methods is offered. Variance reduction in stochastic gradient-based VI is a rich area of research and without those comparisons it is simply not possible to evaluate the performance of the method. The only baseline is a the vanilla MCVI method which is a very elementary baseline and it does not allow the reader to assess the performance of the quantized approach against other variance reduction techniques. Comparison with variance reduction/Rao-Blackwellization methods based on control variates should be included. Furthermore, an experimental comparison with quasi-MC methods should also be included given the strong similarities with the proposed approach. Besides of the lack of relevant baselines, the experiments focus on very simple models. However, the biggest benefit of variance reduction approaches often comes from deep models such as Bayesian neural networks and deep exponential families. I will consider shifting to an accept position to weak accept is these comparisons are included in the rebuttal and incorporated in the camera ready. I will also consider an accept position if an additional more complex experiments with proper baselines is included.

Correctness: The methodology is correct but the lack of baselines make very difficult to properly estimate performance against relevant VI variance reduction alternative.

Clarity: Yes

Relation to Prior Work: The coverage of the related literature is somewhat lacking. More detailed discussion should be given of other variance reduction methods and their relationships with the new approach. In particular, the differences and similarities with control variates methods should be extensively discussed as they are currently dominant in the literature and in applications.

Reproducibility: Yes

Additional Feedback: The method is elegant and it has potential but its usefulness cannot be really assessed without a much stronger experiments section. This paper can definitely turn into an high quality submission if proper attention is given in including relevant baselines and more challenging experiments with deep models.


Review 4

Summary and Contributions: The paper proposes a biased, zero variance estimator for the gradients of the ELBO based on Optimal Voronoi Tesselation of q(z|x). It is shown that this esimator is a lower bound on the ELBO. UPDATE: Based on the authors' response, I'm raising my score to 6, mainly because the general idea is neat. I still have reservations about the quality of the presentation, but hopefully that can be improved for the final version. Also, I meant to ask for a comparison to IWAE not IWAE+quantization.

Strengths: This is a neat idea attacking an important problem and deserves begin explored. The theoretical part seems to check out, although I didn't go into the details.

Weaknesses: The proposed Quantized VI method employs several (20) "samples" (the middle points of the voronoi cells) to estimate the gradients of the ELBO. My main concern is that the ELBO is hardly the best way to make use of several samples. IWAE, or better yet, DReG (see "Doubly Reparameterized Gradient Estimators for Monte Carlo Objectives") would make the experiments much more informative.

Correctness: Yes.

Clarity: Unfortuately, the paper feels rather rushed and draftlike. The content is mostly there, but it is left to the reader to connect the pieces. Spelling could be improved. Sometimes confusing mistakes are made ("gradient-free" is used multiple times instead of "variance-free").

Relation to Prior Work: Yes.

Reproducibility: Yes

Additional Feedback:

[Author Response · NeurIPS 2020]

We thank the reviewers for the valuable comments and suggestions made. We will focus on addressing the main remarks regarding baseline, scalability, complexity and the full batch setting in the following paragraphs.

**Baseline and models.** The reviewers' main concern is the lack of baseline besides MCVI and the absence of a more complex model. We recognize that this gap makes the evaluation of the method difficult. Our main goal was to demonstrate that the use of OQ offers theoretical guarantee and can be efficiently used for inference. Taking into account the suggestions made by all reviewers, the new version includes three baselines: Monte Carlo Variational Inference (MCVI), Quasi Monte Carlo Variational Inference (QMCVI), Randomized Quasi Monte Carlo Variational Inference (RQMCVI). Notably, QVI converges faster on almost all experiments except on the Poisson GLM experiment where similar performance with QMC is observed (Figure 1, second column displays the result for the Forest experiment). In addition, following [7,24] and the suggestions of **reviewers 1,2,3**, we included a more challenging Bayesian Neural Network (BNN) experiment with a larger dataset. The network consists of a Multi Layer Perceptron (30 neurons ReLU activated) with normal prior weights and inverse Gamma hyperprior on mean and variance. The dimension of the latent space is $K = 62$ and regression was performed on the UCI Metro dataset with $L = 48204$ data points (see Figure 1. RQVI procedure led to computational instability). Notably, it exhibits quick convergence for QVI with a bias of $5\%$. Increasing the number of neurons (and thus the posterior dimension) beyond this setup results in a too large bias of $20\%$ in the ELBO estimation. Altogether the experiment section amounts to five methods, three baselines, three models (Bayesian Linear Regression (BLR), Poisson GLM, BNN) and five datasets (Boston, Fires, Life Expect., Frisk and Metro) with learning rate analysis.

Figure 1: ELBO (first row, log scale) and expect gradient norm (second row, log scale) during the optimization procedure for the BNN model (Metro dataset, left) and the BLR model (Forest dataset, right) as function of time. Variance for MC (red area) and QMC (orange area) estimator is obtained by 20 re-run for each experiment.

**Scalability.** Questions were raised by **reviewers 1,2,3** about the scalability of the method with respect to dataset size and dimension. We do not claim that this method is suitable for high dimensional posteriors. We considered it to be the main limitation of the approach as it is for local HPV [24]. For a MC sample size $N$, when considering the $d$-dimensional variational distribution $X^{\Gamma_N,\lambda}$ in place for $X^\lambda$, we introduce a bias in $\mathcal{O}(N^{-\frac{\alpha}{d}})$ for the ELBO estimation. The number of data points $L$ is not a bottle-neck since the complexity associated with computing the MC and QVI estimators are similar (see Eq. 7 for the cubature formula). Active research is underway to reduce bias in higher dimensions [22,26,28].

**Complexity.** To address the question of **Reviewer 1**, the complexity of getting an approximation of the optimal quantizer $X^{\Gamma_N,\lambda}$ is in $\mathcal{O}(N \log N)$ [33] but only needs to be constructed once and can be used throughout the inference since optimality is preserved for the variational family considered. Consequently, the construction of the optimal quantizer is not a limiting factor. It is accurate that the method will not be viable without this property.

**RP gradient in the full batch setting. Reviewers 2,3** pointed out the lack of sufficient discussion about the importance of gradient variance in the full batch setting for the ELBO minimization problem. Gradient variance and CV methods are discussed more thoroughly in [7,24,5] (see L35-L42) and it is an essential issue in stochastic optimization in general. To **reviewer 2**, the gradient variance is displayed in all experiments as red shaded area on even rows (description of gradient variance evolution has been clarified as it can lead to confusion) and is computed on 20 re-reruns.

A relevant point is raised by **reviewer 2** about the full-batch setting. It is true that we consider only the variance associated with sampling from the variational family while in mini-batch sampling, the dominant term would be in $O\left(S^{-1}\right)$ for S-sized batches. We underline that i) it would not exhibit significant variance reduction except on large datasets; ii) even though it would reduce MCVI RP gradient variance, it would also reduce its norm, making it difficult to assess the relative gain for the MCVI method; iii) choosing the batch size $S$ can be difficult, depends on other hyperparameters and is currently beyond our analysis scope. The chosen framework was motivated by extending previous studies [7,24] with full batch RP gradient to deterministic sampling. The new version includes the motivation for the choice of the full batch setting and the comparative performance of control variate and alternative sampling ([7] shows that RQMC outperforms HPV control variate [24] in a similar setting).

**Other comments.** As underlined by **reviewer 2**, the explanation about how to use this method for model checking can be confusing. Put simply, since QVI converges in fewer epochs, we can estimate $\mathcal{L}(\lambda)$ with its quantized counterpart $\widehat{\mathcal{L}}_{OQ}^N(\lambda)$ with a precision given by theorem 1. As pointed out by **Reviewer 4**, we agree that this approach could be better suited for IWAE/DReG/Jackknife VI. However, our derivations rely on the optimal quantizer's technical properties, and it is quite challenging to use it for these gradient estimators (more precisely, it is likely that consistency is not preserved).

[Meta-Review · NeurIPS 2020]

The paper proposes a new optimization method that is based on sampling latent variables. Their approach is based on constructing a optimal Voronoi tessellation that leads to biased but variance free gradient estimates. Strengths: - approach can be implemented in probabillistic programming languages Weaknesses: - comparisons with variance-reduced MC-VI methods are missing